# Enhanced DC Dielectric Properties of Crosslinked Polyethylene Comprehensively Modified by the Grafting of a Multifunctional Voltage Stabilizer

**DOI:** 10.3390/polym16010119

**Published:** 2023-12-29

**Authors:** Peng Li, Xuan Wang, Jin Jin, Xiangxiang Sun, Hui Zhang, Runsheng Zhang

**Affiliations:** Key Laboratory of Engineering Dielectrics and Its Application of Ministry of Education, Harbin University of Science and Technology, Harbin 150080, China; lp_hrbust@126.com (P.L.); emailatjin@163.com (J.J.); sxx495165297@163.com (X.S.); huizhang@hrbust.edu.cn (H.Z.); zrs2583148@163.com (R.Z.)

**Keywords:** crosslinked polyethylene, 1,1′-(oxalylbis(4,1-phenylene))bis(1H-pyrrole-2,5-dione) (BVM), voltage stabilizer, deeply trapped energy level, space charge, breakdown strength

## Abstract

In this paper, a new multifunctional compound, 1,1′-(oxalylbis(4,1-phenylene))bis(1H-pyrrole-2,5-dione) (BVM), is grafted onto crosslinked polyethylene (XLPE) by radical-initiated grafting to play triple roles as a voltage stabilizer, space-charge inhibitor and crosslinking auxiliary and to achieve the purpose of comprehensively enhancing the DC dielectric properties of polymers while decreasing the type and number of additives. By analyzing the DC breakdown field strength, current density and space-charge distribution of the materials at different temperatures, it is demonstrated that BVM grafting can comprehensively and effectively enhance the electrical properties of the materials, with little dependence on temperature. The BVM molecule has two polar groups and an effective molecular structure that acts as a voltage stabilizer, thus enabling the introduction of dense, uniform, deeply trapped energy levels within the material to inhibit the space charge and to capture high-energy electrons to prevent damage to the material structure; however, the two functions do not affect each other. This is also consistent with first-principles calculations and quantum-chemical calculations. Gel content testing shows no effect on polymer crosslinking, even with a 27.8% reduction in the amount of the crosslinking agent di-isopropyl peroxide (DCP), which reduces the damage to the polymer’s electrical resistance caused by the byproducts of DCP decomposition. Therefore, grafting multifunctional BVM compounds to improve the dielectric characteristics of polymers is a viable area of study in the development of high-voltage direct current (HVDC) cable materials.

## 1. Introduction

With the seriousness of environmental pollution and the shortage of fossil energy, green power, meaning clean and renewable power generation, has gradually replaced traditional power generation as the main source of human power for the future [1]. Several major energy systems of green power, such as wind and solar, have the characteristics of being widely distributed and distributed in the reverse direction of the power load center. Therefore, HVDC transmission, which can realize long-distance, large-capacity and low-loss power transmission, has become the main choice for green power transmission [2]. Crosslinked polyethylene (XLPE) has been the mainstream material for insulation in DC cables since the 1960s due to its excellent heat and electrical properties and low cost. However, with the large-scale use of XLPE as a cable insulation material, the role of XLPE in the production, operation and other aspects of many problems and technical difficulties have also gradually appeared. In order to improve the DC dielectric properties of XLPE, researchers usually add polar compounds or voltage stabilizers in the production of cables, but most of these substances are organic small molecule compounds, which not only have a low boiling point and often affect the equipment during the production of gasification, but also have poor compatibility with the XLPE, and it is difficult for these to be dispersed uniformly in the XLPE for a long period of time. In addition, XLPE cable production mostly applies DCP as the crosslinking agent, and the production process produces byproducts such as kukui alcohol and acetophenone, which form air bubbles inside the material. Even if these are separated by heat outgassing, they can also cause structural damage to the internal structure of the XLPE, which increases the power transmission loss and can even severely reduce the life span of the DC cables [3]. Therefore, the study of polymer materials with excellent DC dielectric properties is a current research and development priority.

Current methods to improve the DC dielectric properties of crosslinked polyethylene include enhancing the purity of the material, nanoparticle modification and chemical modification [4]. Nanoparticle modification is the introduction of inorganic nanomaterials containing deeply trapped energy levels into polymers, which, in turn, improves the space-charge-suppression properties of the material, thus improving the DC dielectric properties of the material [5,6]. However, the development potential is limited because of the poor compatibility of inorganic nanoparticles with the polyolefin insulation material, which makes it difficult to disperse and easy for the processing filter to become clogged [7,8]. Chemical modifications include the use of voltage stabilizers and the introduction of functional organic small-molecule compounds to improve the electrical properties of polymers [9,10,11]. Maleic anhydride (MAH) is currently the most commonly used polar compound. Quirke has theoretically investigated the energy distribution of traps generated by physical and chemical defects by means of first-principles electronic structure calculations, which reasonably indicate that polar groups can form carrier deep traps in polymer materials [12,13]. Lee chemically succeeded in grafting maleic anhydride (MAH) onto the macromolecular chain of low-density polyethylene, showing that the anisotropic space-charge accumulation and current density in LDPE-g-MAH can be unambiguously suppressed [9]. Currently, the most commonly used voltage stabilizers are those that capture high-energy electrons [14]. These voltage stabilizers can capture the high-energy electrons in the insulating materials under a strong electric field and reduce the energy of the electrons, thus weakening the influence of the high-energy electrons on the molecular chain and, consequently, improving the breakdown strength of the material. Researchers have extensively explored the mechanism of action of voltage stabilizers through theoretical chemical calculations and found a relationship between the efficiency of voltage stabilizers and their quantum-chemical properties. Jarvid et al. found that the efficiency of voltage stabilizers is closely related to their electron-affinity potentials [15]. Zhang et al. confirmed that efficient voltage stabilizers tend to have high electron-affinity potentials, low ionization potentials and narrow energy gaps [16,17]. In practical applications, since voltage stabilizers are mostly low-molecular-weight aromatic compounds with poor compatibility with polymers, they are prone to migration and precipitation, which will gradually lead to the loss of their initial effect [18]. In order to solve this problem, researchers have carried out many studies and found that graftable voltage stabilizers are the most feasible research direction at present. Dong et al. [19] synthesized a graftable voltage stabilizer that can be grafted onto XLPE molecular chains during crosslinking. It can improve the breakdown strength and inhibit the migration of the voltage stabilizer. Yamano [20] found that when polar groups are found on benzene-ring compounds, deep traps can also be introduced that have the effect of suppressing the space charge. Although both methods improve the DC dielectric properties of polymers to some extent, the effect is one-sided. A voltage stabilizer can significantly improve the DC breakdown strength of the material, but the inhibition effect on the space charge is not obvious enough. However, although functional organic small-molecule compounds have a good effect on the suppression of the space charge and can significantly reduce the current density of materials at high fields, they produce limited improvement in breakdown strength. Although it has been proposed to incorporate both voltage stabilizers and functional organic small-molecule compounds into polymers, the actual result is very unstable. The reason for this is that the two can easily interact or even react chemically in the polymer, thus affecting its performance. Furthermore, adding too many additives to the polymer can lead to an increase in impurities, which can negatively affect its electrical properties. Much research has been conducted on how to reduce the amount of DCP in the production of XLPE without affecting its performance. Yuan et al. [21,22] reduced the amount of DCP by adding the crosslinking additive triallyl isocyanurate (TAIC) in the XLPE production process. The results show that after adding TAIC and reducing the amount of DCP, the thermal elongation, tensile strength and gel content of the material remain essentially unchanged and comply with the relevant industry standards, and the material’s dielectric loss factor drops from 0.00030 to 0.00026, with a slight improvement in electrical performance. These indicate that it is feasible to reduce the amount of DCP by adding TAIC, while the small improvement in the electrical properties is due to the fact that TAIC does not contain functional groups or molecular structures that can significantly improve the electrical properties of the material.

By analyzing the several modification methods mentioned above, we find that although each of these methods has its limitations, their advantages complement each other. Based on this line of thought and the research results obtained so far by our group [23], we selected an aromatic ketone compound, 1,1′-(oxalylbis(4,1-phenylene))bis(1H-pyrrole-2,5-dione) (BVM), containing a multifunctional structure, as shown in Figure 1. The middle part of its molecular structure is the typical functional structure of a voltage stabilizer, and the two sides are functional polar groups, which function as both voltage stabilizers and inhibitors of the space charge. The results of first-principles calculations and quantum-chemical calculations also show that the different functional structures of the BVM all function even when coexisting in a single molecular structure, without interfering with each other to produce negative effects. Meanwhile, the BVM molecule contains two double bonds, which can also act as a crosslinking aid like TAIC, thus reducing the amount of DCP used. In addition, the boiling point of BVM is much higher than the production and operating temperatures of XLPE, avoiding the defects of traditional small-molecule polar compounds that produce decomposition and evaporation to form bubbles. According to the thoughts above, BVM has the triple effects of being a voltage stabilizer, functional polar molecule and crosslinking auxiliary at the same time, which not only reduces the content of polymer additives but also avoids the negative effects brought about by the interactions between different additives, thus achieving the purpose of comprehensively improving the dielectric properties of polymers.

## 2. Experiments and Simulation Calculations

### 2.1. Sample Preparation

LDPE (LD 200 GH, Sinopec Co., Ltd., Jilin, China) is used as a substrate, di-isopropyl benzene peroxide (DCP, Nobel Co., Ltd., AK, Tianjin, China) as a crosslinker, 1,1′-(oxalylbis(4,1-phenylene))bis(1H-pyrrole-2,5-dione) (BVM, BirdoTech, Shanghai, China) as a modifying additive and pentaerythritol ester (Irganox1010, Macklin Co., Ltd., Shanghai, China) as an antioxidant.

In order to accurately reflect the modifying effect of BVM on the DC dielectric properties of the materials, the unmodified XLPE is used as a blank control group for comparative experiments in this paper. The first step is to prepare a homogeneous blend of the two materials. LDPE and 0.3 wt% 1010 are added to a torque rheometer and mixed at 110 °C and 50 rpm for 5 min under experimental conditions. Then, 1.8 wt% DCP is added and the mixing continued for 2 min to produce a homogeneous mixture of raw materials for the preparation of XLPE. LDPE and 0.3 wt% 1010 and 0.3 wt% BVM are added to the torque rheometer and mixed at 200 °C and 50 rpm for 15 min, removed and cooled, and then re-added to the 1.4 wt% DCP in the torque rheometer. This is mixed at 110 °C and 50 rpm for 3 min to produce a homogeneous dispersion of the preparation of the XLPE-g-BVM raw material.

In the second step, XLPE and XLPE-g-BVM are prepared by using free radicals generated by pyrolysis of the peroxides. The two blends are placed in the mold into the plate vulcanizing machine, keeping the temperature at 110 °C. After the material melts, the pressure is increased from 0 to 15 MPa at a rate of 5 MPa every 5 min so that the material is pretreated; after that, the raw material is transferred to another plate vulcanizer while it is still hot, and the heat treatment is kept at a temperature of 175 °C and a pressure of 15 MPa for 30 min to enable the peroxy bonds in the DCP molecule to generate alkyl radicals by a thermal homolytic reaction. Curacil alcohols and polymer radicals are the byproducts of peroxide decomposition that are created when alkyl radicals remove hydrogen atoms from the polymer chain. The BVM molecule’s double bond opens as a result of the free radicals being transferred from the polymer to it, creating a carbon–carbon crosslink with the polymer chain. Subsequently, the free radicals take over the hydrogen atoms in the other polymer chain, forming XLPE-g-BVM and the polymer radicals. In addition, the crosslinking reaction terminates and carbon–carbon crosslinks are created between the two polymer radicals. Kukui alcohol, the unstable byproduct of peroxide decomposition, can break down at high temperatures into acetophenone and methane or styrene and water, together with other decomposition byproducts that adversely impact the material’s dielectric characteristics [24,25]. At the end of the reaction, the material is transferred to a cooling unit and water-cooled to room temperature at 15 MPa. The resulting material is placed in a vacuum oven and heat-degassed at 80 °C for 48 h to eliminate internal stresses and to remove reaction byproducts to obtain XLPE and XLPE-g-BVM. The reaction mechanism is shown in Figure 2:

### 2.2. Characterization and Measurement

In order to test whether the relevant mechanical properties of the modified materials meet the standards, this paper takes GB/T 2951.21-2008 [26] and ASTM D 2765-2011 [27] as the basis and tests the thermal elongation and crosslinking degree of the specimens through the thermal extension test and the gel extraction experiment. To verify whether the BVM molecules are grafted onto the polymer molecular chains, the samples are examined using Fourier-transform infrared (FT-IR) spectroscopy (FT/IR-6100, Jiasco Trading Co., Ltd., Shenyang, China) in the spectral range of 500~4000 cm^−1^ with a scanning resolution of 2 cm^−1^. In this paper, an internationally popular pulsed electroacoustic system (HY-PEA-DPT 01, HeYi Electric Co., Ltd., Shanghai, China) is used to examine the samples by first applying an electric field of 40 kV/mm for 1800 s of polarization and then short-circuiting the samples for 1800 s at temperatures of 40 °C, 60 °C and 80 °C so as to measure the material’s internal space-charge distribution during polarization and short-circuiting. The test material is a 50 mm diameter film sheet with a thickness of 200–300 μm. Conductance currents are measured at temperatures between 40 °C and 80 °C by means of a three-electrode system, and the specimen is a 100 mm diameter, 100 µm thick sheet with aluminum film electrodes on both sides. Ring-shaped protection electrodes (with inner and outer diameters of 54 mm and 76 mm, respectively) surround a disk of measurement electrodes (50 mm in diameter) on one side of the membrane sample and a larger circular electrode with a diameter of 78 mm on the other side for applying high voltages. The measurement protocol is as follows: At 40 °C, 60 °C and 80 °C, the field strength is elevated by a gradient at a rate of 5 kV/mm every 30 min until the field strength reaches 50 kV/mm, and polarization is carried out for 30 min after each elevation in order to measure the stable conductive current, so as to calculate the current density under different field strengths based on the ratio of the conductive current to the specimen area. According to the international standard IEC 60243 [28], using the DC breakdown system with a speed of 1 kV/s uniformly increases the field strength of the DC breakdown test for a specimen of 100 mm and the thickness of a 50–80 μm circular thin film. The specimen and electrodes are submerged in silicone oil during the experiment to avoid any discharge along the surface. The DC breakdown strength is computed by reading the voltage value at the moment of breakdown. The Weibull statistical distribution method is utilized to process the test findings, and the material’s characteristic breakdown field strength is determined by taking the breakdown field strength with a cumulative damage probability of 63.2%.

### 2.3. Molecular Model and Simulation Calculations

The created initial polymer structures are geometrically optimized by total-energy-generalization minimization using a conjugate gradient algorithm, based on first-principles calculations [29,30]. In order to investigate the band-edge features and trap states created by grafting, an electronic structure is computed for the molecular orbitals and electronic density of states [31,32]. First-principles calculations are performed by employing the Materials studio 8.0 software package (Accelrys Inc., v8.0.0.843, San Diego, CA, USA). Detailed schemes and parameters used in the DMol3 calculations are shown in Table 1.

On the basis of the density-functional theory (DFT), full optimization of the ground-state equilibrium geometries of the neutral and ionic states of the investigated molecules has been carried out using the B3LYP generalized approach in the 6-311 + G (d,p) basis group. By using a straightforward positive mode analysis, the computed harmonic vibrational frequencies are shown to have genuine frequencies that correspond to all vibrational modes. This information is used to determine the molecules’ highest and lowest occupied orbitals (HOMO-LUMO), energy gaps (Eg), ionization potentials (IP), and electron affinity potentials (EA). The GAUSSIAN 09 software package is used to perform all the computations on the electronic structures [33].

## 3. Results and Discussion

### 3.1. Testing and Characterization of Crosslinking

In order to verify whether some physical properties of the material are changed after modifying XLPE with BVM, this paper adopts the thermal extension test and gel extraction experiment to check the relevant properties of the material, and the results are shown in Table 2.

By analyzing the data in Table 2, we can see that there is no significant change in the thermal elongation and gel content of XLPE and XLPE-g-BVM, and that some of the physical properties of XLPE modified with BVM remain unchanged. According to GB/T 2951.21-2008 and ASTM D 2765-2011, a thermal elongation of less than 80% is regarded as qualified, and the gel content should be more than 80%, which can be seen in Table 2; XLPE-g-BVM and XLPE are in line with the relevant standards.

### 3.2. Molecular Structure Characterization

In order to verify how the BVM molecules are present in the polymer, the three materials, XLPE, XLPE-g-BVM and LDPE + BVM were chemically characterized by Fourier-transform infrared spectroscopy using XLPE, polyethylene and the BVM physical blends, LDPE and BVM, as a control group in this paper, and the results are shown in Figure 3:

By comparing the infrared spectra of the three materials, we find that, compared with XLPE, XLPE-g-BVM has three new characteristic absorption peaks at 1723 cm^−1^, 1602 cm^−1^ and 1400 cm^−1^, which are the characteristic absorption peaks of the carbonyl group telescopic vibration, the characteristic absorption peak of the ring oscillation of the benzene ring when the carbonyl group is conjugated and the characteristic absorption peak of the telescopic vibration of the C-N bond, which is consistent with the molecular structure of BVM. In addition to the above new absorption peaks, there is also a new absorption peak at 795 cm^−1^, which represents the C=C bond stretching vibration absorption peak. As the materials have been fully degassed at high temperatures, there are basically no free small-molecule compounds inside. We can conclude that BVM is grafted onto the polymer molecular chain by opening the double bond, rather than blending, which also explains the reason for reducing the dosage of DCP without changing the physical properties of the material.

### 3.3. Space-Charge Distributions

The space-charge distributions of XLPE and XLPE-g-BVM at various temperatures from 40~80 °C are tested using PEA [34,35,36,37,38] to analyze the effect of modification on the space-charge suppression ability. The results are shown in Figure 4, Figure 5 and Figure 6.

Figure 4, Figure 5 and Figure 6 show the space-charge distribution results of XLPE and XLPE-g-BVM at 40 °C, 60 °C and 80 °C, respectively. During polarization, the space-charge density inside both materials increases to different degrees with increasing temperature. At 40 °C and 60 °C, the two levels of XLPE are dominated by anisotropic and then isotropic charges, meaning that the origin of the space charge changes quickly from impurity decomposition to electrode injection, and that the peak space-charge densities are all inside the material, which are 2.468 C/m^3^ and 3.821 C/m^3^, respectively. Meanwhile, at 80 °C, the accumulation of a space charge is basically from electrode injection, with a maximum space-charge density of 5.419 C/m^3^; on the contrary, we find that the anisotropic charge near the positive pole of the XLPE-g-BVM consistently dominates throughout the temperature test range, indicating that there is no significant increase in the space charge generated by the electrode injection. The the maximum space-charge density at 40 °C, 60 °C and 80 °C is 1.918 C/m^3^, 1.933 C/m^3^ and 2.431 C/m^3^, respectively, which are smaller than that of XLPE under the same test conditions, while the increase in the space-charge density maximum from 40 °C to 80 °C for XLPE-g-BVM is 26.7%, which is much smaller than the 119.6% increase for XLPE. During the short-circuit process, the space charge inside both XLPE and XLPE-g-BVM starts to dissipate, but the dissipation rate in XLPE-g-BVM is faster, and the residual amount of XLPE-g-BVM space charge at the end of the short-circuit is 0.347 C/m^3^, 0.439 C/m^3^, and 0.465 C/m^3^, respectively, while the residual amount of the XLPE space charge is 0.633 C/m^3^, 0.856 C/m^3^ and 0.969 C/m^3^, respectively.

In summary, the ability of the modified XLPE-g-BVM to suppress the space charge is significantly improved throughout the temperature region measured, especially in suppressing the space charge injected at the electrodes. Its ability to suppress the space charge is less temperature-dependent and more suitable for high-temperature working environments, indicating that a grafted BVM is able to introduce deep carrier traps inside polymers, thus successfully suppressing the build-up of space-charge within the material.

### 3.4. Conductance Current Characteristics

A three-electrode test system is used to test the DC electrical conductive current of the two materials at 40 °C, 60 °C and 80 °C. The final J-E curve of the current density along with the field strength variation curve is obtained, and the results are shown in Figure 7:

As shown in the figure, the current density of the modified XLPE-g-BVM is significantly lower compared with that of XLPE in the temperature range, indicating that BVM has the effect of inhibiting carrier migration. At the same time, the J-E profile of XLPE varies remarkably as a function of temperature; the temperature rises from 40 °C to 80 °C, and its current density at 20 kV/mm, which is the cable’s conventional working field strength, increases by nearly 62 times. Under similar circumstances, XLPE-g-BVM only increases approximately 16-fold, an increase of only one-fourth that of XLPE, indicating that XLPE-g-BVM exhibits significant high-temperature resistance in suppressing carrier migration. This study uses the segmented linear-fitting approach to examine the J-E profile’s properties in order to assess them more precisely. Table 3 presents the slopes of the J-E curves at different stages (k_D_) and the threshold value of the electric field strength at the point of maximum curvature of the curve (E_Ω_).

From Figure 7, we can intuitively see that in the yellow region (Region I), the k_D_ of the J-E curves of the two materials is smaller, and from Table 3, we can see that its value is about 1. J is proportional to E, which is in line with the characteristics of Ohm’s law, so this region is also known as Ohm’s conductivity zone. As the temperature and field strength increase, the J-E curve moves from the ohmic region into the trap-action region (the pink region (Region II) in the figure), which is marked by the appearance of the threshold field strength E_Ω_. In this region, the kinetic energy of the holes and electrons emitted from the electrodes is able to cross the potential barrier, leading to a sharp increase in the electrode-injected charge density and a large increase in the conductance current with the amplitude of the change in field strength, causing the J-E to show a non-linear relationship in this region, and the k_D_ in this region is also increased to about 3.5. The law of change in current density with field strength in this region is in accordance with the theory of the space-charge limiting current (SCLC), so this region is also known as the SCLC region. Analyzing the data in Table 3, the threshold field strengths of XLPE-g-BVM at 40 °C, 60 °C and 80 °C are 35 kV/mm, 25 kV/mm and 15 kV/mm, while XLPE can detect the threshold field strengths only at 40 °C, and the threshold field strengths are already less than the detection range at 60 °C and 80 °C. Compared with XLPE, the threshold field strength between the XLPE-g-BVM ohmic region and the trap region of action is increased across the board, which is a reflection of its enhanced role in suppressing carrier migration. With the further increase in temperature and field strength, the maximum curvature point of the J-E curve of XLPE appears again, after which the curve’s k_D_ also drops to about 1 (i.e., the blue region in the figure), which means that the carrier speed within the material already meets the maximum tolerance limit of the material, and cannot continue to be enlarged, which also marks the entry of the J-E curve into the third stage (Region III). In this region, although the conductive current growth rate slows down, it is still easy to generate the cable internal electric field reversal phenomenon, which seriously affects the performance of the cable, so this phenomenon should be avoided. The main reason for the generation of electric field inversion is that the conductivity properties of the material are strongly affected by temperature, while the J-E curve of the modified XLPE-g-BVM did not enter the third period within the measurable range, indicating that its inhibition of carrier migration is less temperature-dependent and more suitable for high-temperature operation.

Based on the above analysis, we can conclude that the threshold field strength to distinguish the ohmic region from the trap region is positively correlated with the ability of the dielectric to trap carriers, and the higher the threshold field strength at the same temperature, the stronger the ability of the material to inhibit the migration of carriers. The threshold field strength between Region II and Region III is inversely correlated with the material’s sensitivity to temperature. Region II and Region III are inversely related to the temperature sensitivity of the material; the higher the threshold field strength, the smaller the temperature dependence of the conductivity of the material, and the stronger the insulation reliability. Both threshold field strengths of the modified XLPE-g-BVM become larger, indicating its effective and reliable inhibition of carrier migration. Analyzing the reason, the BVM molecule includes polar groups that can lead to the introduction of many deep-trap energy levels within the polymer, thus inhibiting the migration of carriers. The introduction of many deep-trap energy levels, only when there is a large increase in the temperature, makes it possible for the electrons to obtain sufficient power to jump across the potential barriers, so its dependence on the temperature is reduced.

In order to investigate whether grafted BVM can introduce deep-trap energy levels within the polymer, the electronic energy density of states and the atomic projected density of states of XLPE-g-BVM and XLPE are calculated according to first principles in this paper. The results are shown in Figure 8.

Figure 8 shows the total density of states for XLPE-g-BVM and XLPE and the projected density of states for carbonyl and aniline. The bandgap interstates between the conduction and valence bands act as charge-trapping sites. As can be seen from Figure 8, there are essentially no charge-bound states between the bandgaps of XLPE, while XLPE-g-BVM still has an abundance of bound states. The projected density of states of carbonyls and anilines shows that carbonyls and anilines can introduce hole traps near the conduction band and electron traps near the valence band inside the polymer, and their functions can be superimposed on each other rather than weakening each other. For example, in Figure 8, the hole trap with an energy level of 1.5 eV is the deep-energy-level hole trap generated by carbonyls and anilines, which is able to capture holes emitted from electrodes more strongly and has less temperature dependence. The deep-energy-level electron traps with energy levels of 2.7 eV, 2.1 eV and 1.6 eV on the conduction-band side are also the same. In addition, BVM also introduces two composite electron centers in the middle of the conduction and valence bands, which are capable of trapping holes and electrons at the same time. This calculation result also matches the previous experimental data, verifying the previous conclusion.

### 3.5. DC Breakdown Strength

The Weibull distribution of the DC breakdown strength of XLPE and XLPE-g-BVM at 40–80 °C is shown in Figure 9. As can be seen from the figure, the DC breakdown strength of both materials decreases with increasing temperature, but the DC breakdown strength of XLPE-g-BVM is always higher than that of XLPE, which indicates that grafting BVM improves the material’s electrical resistance characteristics. The characteristic breakdown field strength can be described in terms of the field strength at a 63.2% breakdown probability, which is obtained directly from the scale and position parameters of the fitted Weibull distribution; these results are shown in Table 4. As can be seen in Table 4, the characteristic breakdown field strength of XLPE-g-BVM reaches 436.9 kV/mm, 382.0 kV/mm and 247.8 kV/mm at 40 °C, 60 °C and 80 °C, respectively, which is an increase of 40.4%, 40.9% and 50.4% compared with XLPE. From this, we can see that BVM improves the electrical resistance with the increase in temperature and makes it stronger; the reason for this is that at 80 °C, the DC breakdown field strength of XLPE drops drastically. At high temperatures, due to the lack of deep-trap energy levels for charge trapping within the XLPE, the kinetic energy of the holes and electrons emitted from the polar plate is able to cross the potential barriers, leading to a sharp increase in the electrode-injected charge density, which is more likely to lead to the decomposition of the internal structure of the material and the generation of distorted electric fields, and ultimately lead to the occurrence of breakdown.

We can infer whether the molecules function as voltage stabilizers based on quantum calculations. In order to determine the principle of why BVM improves the DC breakdown strength, we calculated the quantum-chemical properties of XLPE and BVM on the basis of the density functional theory (DFT), and the results are shown in Table 5. XLPE molecules have a negative electron-affinity potential while BVM molecules have a positive electron-affinity potential. When electrons are accelerated under the electric field in the polymer, they are readily attracted to the BVM molecule and, therefore, have a high probability of attacking the BVM molecule. When the energetic electrons contact the BVM, energy transfer occurs, since the HOMO-LUMO energy gap (Eg) of the BVM molecule is only 3.50 eV, which is lower than the ionization potential (IP) of the BVM molecule and the XLPE molecule (7.84 eV and 9.41 eV, respectively). Therefore, in the case of an inelastic collision between the electrons and the BVM molecule, the BVM molecule absorbs some energy within the 3.50–7.31 eV range, completing the excitation process and releasing the energy in the form of relatively harmless heat. After releasing this energy, the BVM molecule reverts to the ground state and continues to repeat the above steps, thus forming a cycle.

## 4. Conclusions

In this paper, based on the multifunctionalization of the modifier and the relevant research results obtained by the project team, a new multifunctional compound, BVM, is screened and successfully grafted onto the polyethylene molecular chain. Thermal elongation tests and gel-content tests demonstrate that, even with a 27.8% reduction in the amount of DCP used, XLPE-g-BVM is guaranteed to have almost the same degree of crosslinking as XLPE. By organizing the experimental data of the breakdown field strength, conductive current and space-charge distribution at 40–80 °C, it is confirmed that BVM molecules have the dual effects of capturing high-energy electrons and suppressing the space charge. Due to its unique molecular structure, BVM functions as a voltage stabilizer and crosslinking auxiliary and introduces deep-trapping polar groups, which can trap high-energy electrons to protect the integrity of the internal structure, introduce deep-trapping energy levels to suppress the accumulation of the space charge, and reduce the negative impact of the byproducts of the breakdown of DCP on the electrical properties of materials, thus comprehensively improving the DC dielectric properties of XLPE. The results of first-principles calculations and quantum-chemical calculations also confirm the experimental results and reveal the reasons for their versatility: The functions of inhibiting the space charge and capturing high-energy electrons belong to different functional groups in the BVM molecule, which have little to do with the overall structure of the molecule, and the corresponding functional groups are electronegative functional groups, which are relatively similar in nature, and therefore do not interact with each other to weaken their functions. This paper presents a novel approach for realizing the improvement of DC dielectric properties of DC high-voltage insulating materials, which will have practical applications in the production of high-voltage DC cables.

## Figures and Tables

**Figure 1 polymers-16-00119-f001:**
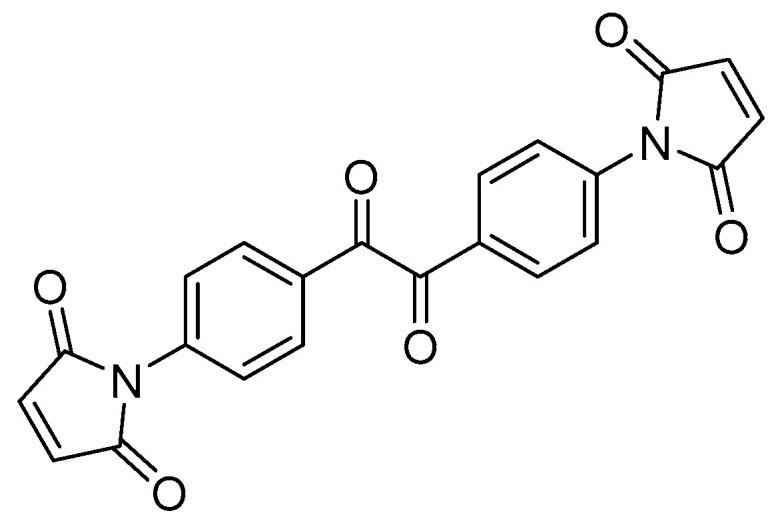
Molecular structure of BVM.

**Figure 2 polymers-16-00119-f002:**
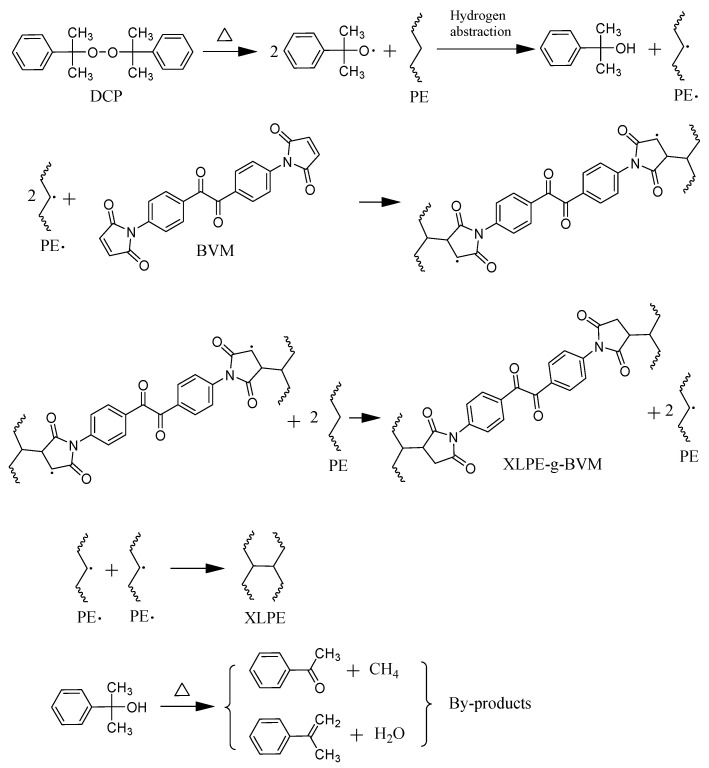
Schematic diagram of the BVM-grafted PE and crosslinking reactions initiated by free radicals.

**Figure 3 polymers-16-00119-f003:**
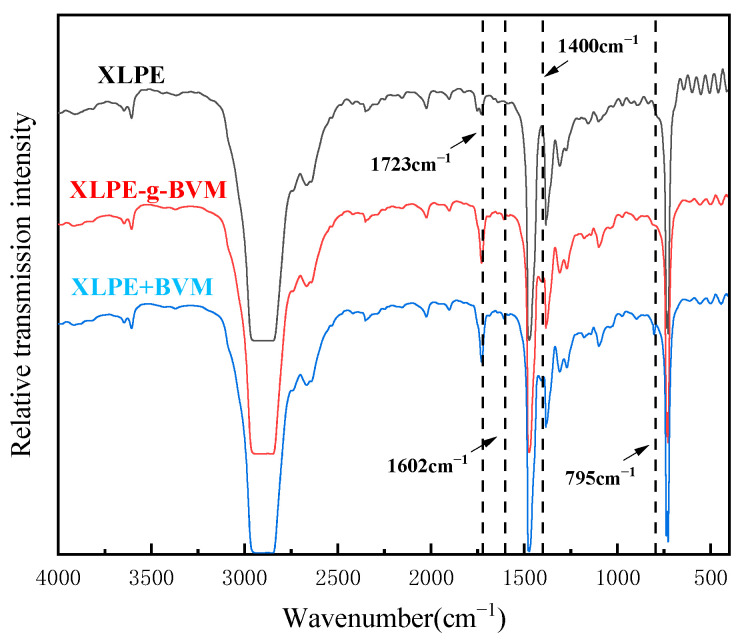
Infrared transmission spectra of XLPE, XLPE-g-BVM, and LDPE + BVM.

**Figure 4 polymers-16-00119-f004:**
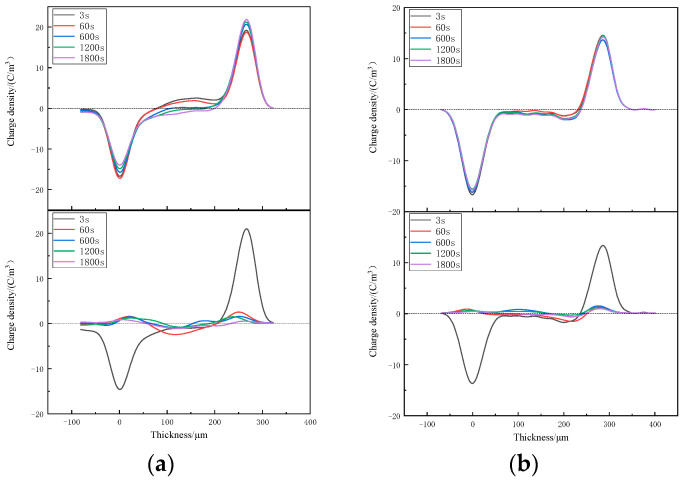
Space-charge distribution in (**a**) XLPE and (**b**) XLPE-g-BVM at 40 °C under 40 kV/mm field strength (top panel) and short circuit (below panel).

**Figure 5 polymers-16-00119-f005:**
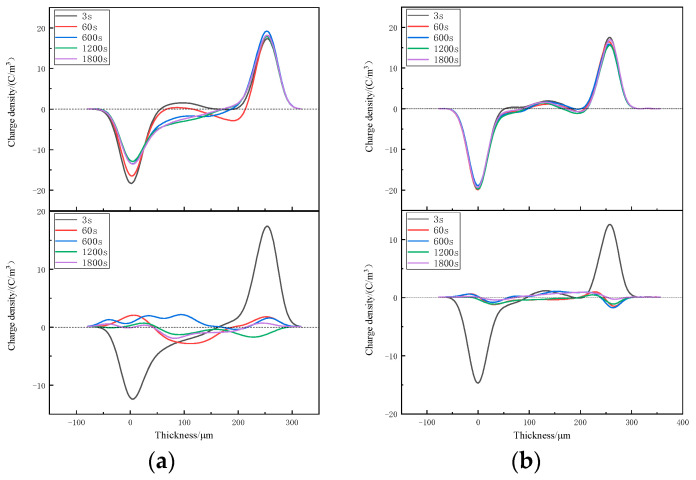
Space-charge distribution in (**a**) XLPE and (**b**) XLPE-g-BVM at 60 °C under a 40 kV/mm field strength (top panel) and in short-circuit conditions (below panel).

**Figure 6 polymers-16-00119-f006:**
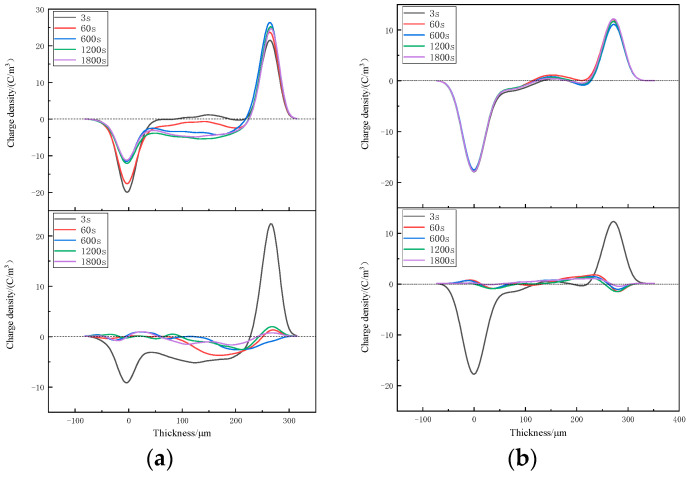
Space-charge distribution in (**a**) XLPE and (**b**) XLPE-g-BVM at 80 °C under a 40 kV/mm field strength (top panel) and short-circuit conditions (below panel).

**Figure 7 polymers-16-00119-f007:**
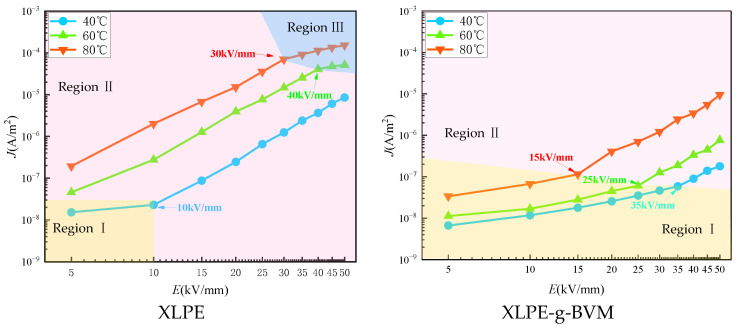
Conductivity properties of J-E varying curves for XLPE and XLPE-g-BVM.

**Figure 8 polymers-16-00119-f008:**
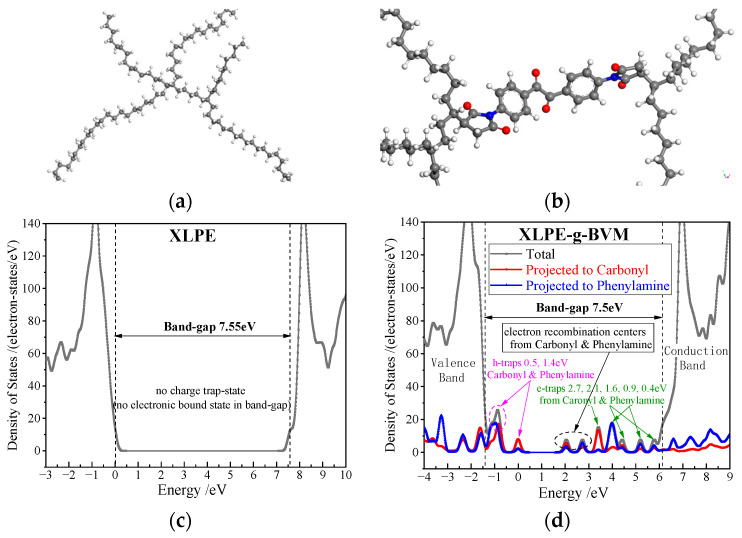
(**a**,**b**) Schematics of XLPE and XLPE-g-BVM with gray, white, red and blue spherules identifying carbon, hydrogen, oxygen and nitrogen atoms respectively; (**c**,**d**) Density of states (DOS) of XLPE and XLPE-g-BVM from first-principles calculations.

**Figure 9 polymers-16-00119-f009:**
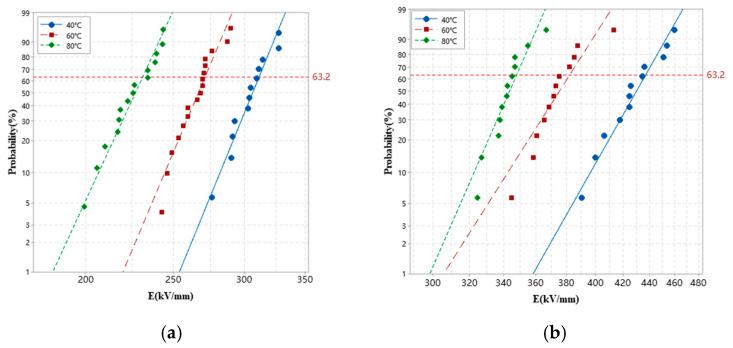
Dielectric breakdown strength (DBS) statistics fitted with the Weibull distribution for XLPE and XLPE-g-BVM at a temperature of 40–80 °C. (**a**) XLPE; (**b**) XLPE-g-BVM.

**Table 1 polymers-16-00119-t001:** Schemes and parameters adopted in the first-principles calculations by DMol3.

Electronic Hamiltonian	Scheme	Condition and Parameter
Exchange-correlation energy	Meta-generalized gradient approximation	M11-L [29]
Integration accuracy		2000 grid points/atom
	Tolerance	1 × 10^−6^ eV/atom
SCF	Multipolar expansion	Octupole
	Charge-density mixing	Charge = 0.3, DIIS = 5
Core treatment	All-Electron	
Numerical basis set	DNP	Basis file 4.4
Orbital cutoff	Global	5.0 Å

**Table 2 polymers-16-00119-t002:** Heat extension rate and the gel content of two materials.

	Heat Extension Rate (%)	Gel Content (%)
XLPE	35	85.3
XLPE-g-BVM	33	86.1

**Table 3 polymers-16-00119-t003:** The current density slope (k_D_) and threshold electrical field strength (E_Ω_) at different temperatures.

	40 °C				60 °C				80 °C			
	k_D_ (A/(kVmm))			E_Ω_ (kV/mm)	k_D_ (A/(kVmm))			E_Ω_ (kV/mm)	k_D_ (A/(kVmm))			E_Ω_ (kV/mm)
	I	II	III		I	II	III		I	II	III	
XLPE	0.88	3.75		10		3.33	1.01	40		3.23	1.52	30
XLPE-g-BVM	1.09	3.49		35	1.06	3.53		25	1.1	3.5		15

**Table 4 polymers-16-00119-t004:** The characteristic 63.2% DBS of the Weibull distribution fitted in a 95% confidence interval at different temperatures (kV/mm).

	40 °C	60 °C	80 °C
XLPE	311.1	271.1	231.2
XLPE-g-BVM	436.9	382.0	347.8

**Table 5 polymers-16-00119-t005:** Quantum-chemical characteristics of XLPE molecules and XLPE-g-BVM molecules.

	Eg (eV)	IP (eV)	EA (eV)
XLPE	8.38	9.41	−1.09
XLPE-g-BVM	3.50	7.84	2.12

## Data Availability

The data presented in this study are available on request from the corresponding author.

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
