# Peer review of "Enhanced DC Dielectric Properties of Crosslinked Polyethylene Comprehensively Modified by the Grafting of a Multifunctional Voltage Stabilizer"

_polymers, 2023, doi:10.3390/polym16010119_

Round 1

Reviewer 1 Report (Previous Reviewer 2)

Comments and Suggestions for Authors

Author Response

Reviewer 2 Report (Previous Reviewer 1)

Comments and Suggestions for Authors

The authors introduce a multi-functional compound, BVM, grafter onto polyethylene chains, enhancing cross-linking in XLPE with reduced DCP. BVM demonstrates dual effects of capturing high energy electrons and suppressing space charge, improving DC dielectric properties. Its unique molecular architecture acts as a voltage stabilizer and crosslinking auxiliary, holding practical promise for high-voltage DC Cable production. 

Following are some of my suggestions/questions:

Line 34: This long sentence doesn’t make sense to me. Please break it down.

Line 96- 116: Rephrase this paragraph to highlight the work done in your paper.

Line 136: What is the dye used?

Line 137: approximately how long did it take for the material to melt?

Comments on the Quality of English Language

Minor editing required

Author Response

This manuscript is a resubmission of an earlier submission. The following is a list of the peer review reports and author responses from that submission.

Round 1

Reviewer 1 Report

Comments and Suggestions for Authors

Even though the work is interesting, the authors have used AI to write the article.

Reviewer 2 Report

Comments and Suggestions for Authors

Comments on the Quality of English Language

Manuscript should be thoroughly checked for sentence correction. There are many incomplete sentences.